# Cost Analysis in Online Teaching Using an Activity Map

**Nuria Segovia-García** * and **Ester Martín-Caro**

School of Economic and Administrative Sciences, Corporación Universitaria de Asturias,
Bogotá 110221, Colombia; emartincaro@iep.edu.es
* Correspondence: tecnologia.ns@asturias.edu.co

**Abstract:** Virtual education has been associated, on some occasions, with economic models of scale that strengthen the mercantilist vision of teaching, thereby raising questions on the quality of the service. Some studies on educational costs associated with this modality simplify the analysis based on individual elements, such as the production of courses or the technological infrastructure, offering an incomplete and biased perspective. This study proposes to carry out a complete analysis of the costs of a virtual university using an Activity-Based Costing (ABC) model to determine the percentage of the budget consumed by each of the activities involved in the training process. The results show the significant influence that instructional design and teaching activities have on virtual modalities (62.5% of the set of activities), thus highlighting the importance of human resources in maintaining educational modalities mediated by technology. Understanding the importance of human capital and its contribution to offering a quality and innovative educational service is fundamental to consolidate a model that can improve access to education for populations with major difficulties.

**Keywords:** educational technology; educational costs; educational planning; educational population; higher education

## 1. Introduction

The virtual modality is becoming a viable alternative to reinforce attention in tertiary education and to overcome problems of coverage and accessibility, which is an insurmountable barrier in many educational systems, especially for certain vulnerable groups who, due to their economic or geographical limitations, have major problems [1]. The number of both public-owned and private universities that have joined the virtual format to complement face-to-face training [2] in recent times is significant; a trend that has accelerated in recent years not only due to the COVID pandemic, forcing many institutions to adopt technology as a means of transmitting knowledge [3,4], but also due to the unstoppable technological progress that has allowed the adoption of 4.0 technologies such as Machine Learning (ML) or Artificial Intelligence (AI) which increase the quality of virtual education by allowing greater personalization of the educational response [5,6].

This boom in online training has also contributed to the growth and expansion of an important industry of technological tools and products that provide added value to training actions. These include Learning Management Systems (LMS), which between 2021 and 2028 is expected to achieve robust growth [7], cost-feasible videoconferencing tools which provide a wide range of benefits to the educational institutions [8], the recent proctoring systems that have become necessary tools to guarantee the evaluation processes in online education [9,10], and all of these together facilitate the development of quality, enriched and ubiquitous learning environments [11].

Among the varied possibilities offered by the entry of technologies into education, some studies point out that an advantage of the virtual modality is the possibility of saving the costs of learning management without sacrificing the capacity to admit students [12] as it is based on a strong initial investment in the production of courses that is amortized with the increase in student ratio [13]. However, understanding the long-term efficiency

gain in its delivery model erroneously simplifies the didactic characteristics and training opportunities of this training modality, because there is no evidence proving that e-learning is a more profitable modality than others, but it is perceptible that, when it is executed at a high level, the development costs become significant [14].

### 1.1. Virtual Modality Requirements

The virtual modality has incorporated important changes in the educational sector due to the transformation caused by technology in educational spaces, in the resources used and in the ways of interacting. That is why, elements such as the classroom, which constitutes a valuable element in learning, have undergone a conversion towards the virtual mode which now requires having an adequate technological infrastructure and establishing design and management protocols that are different from those used in face-to-face modality.

In the sections that follow, we will provide a description of the infrastructure, resources and necessary profiles that are required to make the virtual modality effective in optimal quality conditions.

#### 1.1.1. Technological Infrastructure and Resources

Launching online programs within Higher Education Institutions (HEIs) entails a work process in which the tools, resources, and equipment necessary for the development of the didactic methodology are clearly defined. Osadcha et al. [15] indicate that, in order to correctly implement the methodology, it is necessary to take into account the trends in the development of ICT and its influence on learning, as well as supervise and update software and hardware, and develop simple user interfaces.

The development of e-learning is linked to the use of the Internet and the appearance of the first training platforms (LMS) that initially emerged as spaces for the exchange and review of content, and have now evolved into social models, particularly with the introduction of web 2.0 and social networks [16]. Currently, e-learning has evolved to become a comprehensive model that seeks to offer a personalized response adapted to the needs of each student, thanks to the adoption of 4.0 technology [5,6].

Taking this evolution into account, authors such as Marunevich et al. [17] or Osadcha et al. [15] indicate that the most used tools in virtual mode are social networks, LMS, videoconferences, messaging tools, and course production tools, such as Articulate 360 or Adapt. However, diversity of the tools, resources and spaces that can be used in this formative modality is so wide that it has allowed the development of a highly varied technological ecosystem where it is feasible for different user profiles to interact, generating multiple experiences and learning situations. Ref. [18] states that there is room for other emerging technologies, such as artificial intelligence, augmented reality or robotization, which will presumably change the way universities work in the coming years [19,20].

Currently, one of the first decisions that HEIs must make when they develop their programs in virtual modality is the choice of the technological infrastructure with which they are going to work, considering the LMS, the hosting and distribution system, equipment and the necessary human resources. It is necessary to note that this decision is not easy due to the wide array of options available in the technological market; for example, there are around 500 LMS, with very different services, license, and pricing proposals, depending on whether the solution they offer is based on owner or free software [21].

There are certain open source LMS, such as Moodle, Sakay or Chamilo, which have no license cost, and there are proprietary or commercial solutions, such as Blackboard or Canvas, that have a license cost determined by the company developer where you pay for using the service [22]. In addition to the types of LMS license, there are other elements that determine the cost of the service of these tools, such as the accommodation and delivery of the service, equipment, infrastructure, and personnel that HEIs must have. In this sense, the diversity of available options ranges from the cheapest, consisting of downloading and installing an open source LMS on your own or third-party servers, to the possibility of

outsourcing the service to specialized IT companies with different installation alternatives, accommodation, and maintenance of the platform.

All the options described above are viable to support the infrastructure of an LMS platform in an HEI. However, the option of hosting the system on your own or third-party servers carries a risk in terms of information security, cyber-attacks, or identity theft of users [23]. In some way, the option of hiring a specialized external service increases the cost of the LMS but guarantees the security of the entire learning management system. Within this service outsourcing option, there are solutions that offer infrastructure as a service (IaaS), which guarantee some level of quality performance of the platform, avoiding failures in global services, due to the amount of redundant resources used and supporting tasks of updating, security and monitoring of servers, maintenance of operating systems, databases, balancers and management of backup, and snapshot services, among others [24]. There are also service delivery solutions, such as those offered by Canvas, Blackboard or even MoodleCloud, which, with though have a higher license cost than the previous options, allow the delivery of software as a service (SaaS), that is, through the Internet, thus avoiding the costs of equipment maintenance or updates of systems [22].

In addition to the technological infrastructure, when choosing an LMS, it is important to consider its characteristics and functionalities, along with the didactic methodology for training development. This will determine whether the HEIs need to invest in external tools to enrich the system, in addition to the technological infrastructure. Some of the tools that enrich the educational value are those that provide videoconference services, such as Zoom, Cisco Webex, and Microsoft Teams, which, when integrated into the LMS, increase their possibilities of interaction [17,25–27]. Some other tools contributing to improve students' approach to knowledge and interaction, are repositories or virtual libraries, laboratories, or simulators, among others [28]. In addition, there are other tools which addresses the challenges that the digital transformation is generating in HEIs, such as collusion problems, that is, copying of work between students and plagiarism [29], which, far from being solved with technology, can be minimized with anti-fraud tools, such as SafeAssign from Blackboard, [30] or by integrating software into the LMS that allows to detect plagiarism in assignments and deliveries; there are also other solutions, such as proctoring tools, Agile, which can avoid identity theft in online exams.

### 1.1.2. Production of Materials for Virtual Training

The production of materials for virtual training is another element that HEIs must take into account when deciding their cost management budgets. The design of materials for online training to support teaching requires a methodology that defines the rules for their design, implementation, and evaluation. This is required to obtain materials that encourage active learning [31,32] and guarantee motivation and student satisfaction [33].

In this process of creating visualization and support materials for virtual training, different tools come into play through which it is possible to design courses in video, audio, PDF or presentation format, and these tools also aid in the design of reusable digital resources with a pedagogical purpose, such as Learning Objects (LO), that are based on the SCORM model (Sharable Content Object Reference Model) [34]. The development of OA represents a significant production cost for institutions due to the technical difficulty involved in the use of certain editing software [34] and the cost of tools used for this, such as Adobe Captivate or Articulate 360 [35–37]. There are open-source editing tools, such as Learninga-pps.org, Exelearning.net, app.Lumi.education, via which it is also possible to create, edit, view, and share interactive content without incurring license fee [38,39].

An important part of the design of these LOs is the production of quality audiovisual material that increases students' motivation, satisfaction, and perception of the material, as stated in Mayer's multimedia learning theory [40]. Within this audiovisual material, it is necessary to differentiate between the videos generated through sessions or synchronous meetings with conference software, such as Zoom, Meet, and Jitsi, aimed at the transmission of knowledge, from those educational videos that have been produced based on certain

planning and structure, following specific design rules and whose purpose, among others, is to capture attention, raise awareness, express, reflect or encourage collaboration. Curating such educational videos involve a greater effort in its pre-production, production, and post-production phases. To carry out these steps, it is necessary to have different materials and equipment, such as cameras, microphones, lights, etc., as well as the type of software that facilitates production and post-production, for example, Adobe Premiere, Camtasia or Sony Vegas, which are widely used in the educational field [41].

The design alternatives for virtual educational materials are extensive, as are the tools used, from text editors (for example, Dreamweaver, Atom, VS Code or Sublime Text), image editors (Photoshop, Illustrator, GIMP, to name a few examples), sound editors (Audacity and Ocenaudio, among others), to web services for designing infographics, graphics, mental maps or presentations, such as Canva, Mindomo or Genial that have plans for free or paid access [38,39].

### 1.1.3. Professional Profiles in Online Teaching

Virtual training is carried out in settings different from the traditional ones and consequently requires different professional profiles with digital skills that allow them to carry out their functions optimally.

The professional profiles involved in an e-learning training strategy can be differentiated between the more technical ones associated with the maintenance of the technological infrastructure and the development of resources and materials, academic and management profiles, monitoring, and accompaniment of the student. However, fundamental in these teams is the need for them to be trained to carry out coordinated work, combining the technical and pedagogical part to generate a methodology that is replicable and adaptable, complying with the provisions of the ISO/IEC 19796-1 standard for management, insurance, and metrics of quality in virtual education [42,43].

The type of technological infrastructure used by the HEIs will determine the type of specialists that are needed in the technological part. The HEIs that are fully responsible for the installation and maintenance of the training platform, monitoring its performance, updating versions and patches, developing and installing customizations and extensions of their own or those of third parties, among other functions, want IT specialists [44], while those that have delegated this part to specialized external companies assume this cost within the service provided.

An essential profile in this training modality is the instructional designer who will establish himself/herself as a link between the technical and academic part, developing the guidelines, content sequencing and activity planning so that students can correctly interact with their materials and achieve the proposed objectives [45–47]. The instructional designers will work in coordination with technical profiles in charge of configuring the virtual learning environments so that the interaction within these spaces is optimal and adheres to the methodology established by the IES [43] and also work with the specialists in the production of materials and resources associated with e-learning, such as graphics, web and audiovisual designers who, based on their knowledge of the rules of multimedia and audiovisual language, will be able to build attractive teaching materials and resources.

On the other hand, in the academic part, instructional designers are required to interact with specialists of different subjects in charge of preparing the content of the course that will later be structured in a didactic way by the instructional designers, passing through the teachers, as well as by professors in charge of accompanying and guiding student during the development of the training activity or even by the virtual tutors or facilitators who establish communication bridges between the institution and the students, and accompany the students and teachers in order to help them solve any issue related to the platform or any administrative and academic procedures. These profiles have become fundamental actors in the virtual modality whose role transcends that of the instructor to become a guide and facilitator [48,49].

These teams mentioned above will also be in charge of making proposals for training and continuous updating in the different processes of the educational project that guarantee the development of materials and resources with high pedagogical value based on the principles of Universal Learning Design (DUA). [42]. These constant reviews and the updating of the materials are fundamental to guarantee the satisfaction and perception of quality for those students who seek virtual training to have updated, quality, employable and relevant content, and resources [33–50].

The digital footprint that is generated in the training platforms, along with the wealth of information, demand the inclusion of professional profiles specialized in data analysis who work with this raw data and transform it into useful information to make improvements in strategic planning and decision-making for the training programs and actions while being established as strategic profiles to add value and raise the quality of the virtual model [51,52]. Within the HEIs, having data analysis teams that promote the improvement of other teams associated with the provision of a quality service, such as the mentoring/academic tutoring team, the team in charge of registration processes, and academic/administrative control or the university welfare service, become essential.

The discussion so far in this paper allows us to notice the multiplicity of profiles, skills, processes, and infrastructure that are necessary to develop an online methodology in quality conditions, capable of venturing into the virtual modality, which is certainly not cheaper than others [14]. A striking fact about this is that there are only a few national and international studies that analyze the costs of virtual universities generated by activities and tasks based on appropriate analysis methodologies [53,54].

### *1.2. Scope of Work*

The objective of this study will be, on one hand, to carry out an analysis of the technological infrastructure, resources, tools, and human capital that an HEI must use to offer quality online training services. On the other hand, the objective is to determine the activities that are consuming these resources based on the design of an activity map that will serve to identify and visualize the processes or activities that take place within the different departments of the institution. The activity map shows the interactions between the different activities and processes and helps to identify the costs associated with each activity.

The study will take as reference a higher education institution with a long trajectory in online education and whose methodology is 100% virtual. Due to the characteristics of this institution, the situation created by the COVID-19 pandemic will not be considered as a variable in the study. The current scope of work will focus on analyzing the technological infrastructure and the resources, tools, and human capital required once the HEIs have defined and developed their study plans.

The knowledge of these aspects will allow us to understand the influence that technology and virtualization processes have on a virtual institution and compare them with other costs related to teaching functions and student monitoring, which will improve the understanding of costs incurred while utilizing e-learning modes. On the other hand, this study has used an analysis unit based on the percentage of costs assigned to each activity, instead of using monetary units, which increases the possibility of generalizing adequate results, thereby avoiding limitations established by the currency or the economic context of a specific country.

## 2. Materials and Methods

To achieve the objectives proposed in this study, we collaborated with a Colombian Higher Education Institution, which, for a long time, has been offering undergraduate and postgraduate programs in virtual modality.

This analysis has been temporarily limited to the period from January to December 2021, coinciding with a moment of growth of the HEI. At this time, four undergraduate programs and three postgraduate programs are taught entirely in a virtual mode. In

addition to this, four new programs have been incorporated into the institution's training offer, one undergraduate and three postgraduate courses, all of which again must be developed and implemented in virtual format.

The methodology adopted for the work is that of ABC (Activity-Based Costing), which is based on the general idea that the consumption of costs and resources is not generated by the products, but by the performance of certain activities. Based on this premise, cost is understood as the sum of goods and services necessary to produce the resources and materials associated with virtual courses, as well as the sum of goods and services necessary to provide quality virtual training for students. In this sense, only the activities related to the training process of the students have been taken into account, not taking into consideration those related to the commercial process of attracting students, such as the works of the admissions department or the financial management department.

The procedure that has been followed to analyze the costs of each activity has been the following:

1. Contextualization of the HEI and its educational offer.
2. Data collection was based on interviews with different managers of the following departments: (a) technology; (b) academic; (c) academic registration and control; (d) business intelligence; (e) academic mentoring; and (f) university well-being. These departments were chosen as their specific inputs are specifically important.
3. Grouping of activities in similar units and design of the HEI activity map.
4. Determination of the inducers (cost driver) of resources.
5. In coordination with the financial director of the HEI and those responsible for each department, there will be an analysis of the functional costs that helps in determining the percentage of cost of each activity within the institutional budget.

## 3. Results

The HEI included in our study offers five four-year undergraduate, constituting an average of 130 credits and six one-year postgraduate programs comprising 26 credits. Three of the undergraduate courses have been taught since 2014, while the other two have been launched after their approvals in 2019 and 2021, respectively. The same happened with the other three specializations which have been approved in 2021, at which time the program begins to be implemented and taught.

During the year 2021, an average of 4200 students were enrolled in the programs, of which 75% are undergraduate students and the remaining 25% are postgraduate students.

For the development of the analysis, an orderly process has been followed, starting from (1) the identification of all the resources that consume (input) (Table 1); (2) the elaboration of a map of activities (Table 2); (3) the determination of the inductors (cost driver) of resources (Table 3); and (4) the distribution of costs in activities (Figure 1). The complete process is described below in detail.

### 3.1. Definition of the Inputs

The HEI has a technological solution made up of three interconnected systems through APIs: a customer relationship management that allows receiving student registrations and managing them; a student relationship management that orders the training itineraries and enrolls students in each one of them, in addition to serving as an instrument of institutional and academic communication, facilitating access to different services (financial, administrative, communication, among others); and finally, an LMS.

The building of the technological infrastructure, together with the complete administration of the servers (updates, security, monitoring and optimization), and the corrective, preventive and evolutionary maintenance of the solution are outsourced through an IT service provider. For the period corresponding to the year 2021, the solution contracted by the HEI is made up of 250 GB storage and 11,000 users connected to the annual course. It must be taken into account that the solution is scalable and that, if the contracted volumes were exceeded, it would scale to the next contract tranche that corresponds to it.

Within this technological solution, the provider includes a gateway for sending up to 5000 emails per month and the HEI has expanded this service with a complete email marketing solution that allows sending 100,000 transactional emails per month through the platform.

Along with the above, the system is completed with other external solutions:

1. Integration of a module for webinars and video collaboration from the Zoom tool that consists of ten simultaneous rooms of 100 students, one room with a capacity of 300 users and one room with a capacity of up to 1000 simultaneous users, all of them without a duration limit and with an unlimited number of rooms for virtual tutorials of around 100 participants with a maximum duration of 40 min. The solution comes complete with a space of up to 500 GB to host and broadcast webinar recordings.
2. Integration of a module for surveillance of exams or proctoring, licensed with Smowl, which allows to verify the identity of students through facial biometric recognition and monitor all students' computer activity during the course of the exam. For the 2021 period, a total of 1500 exams have been contracted.
3. Video streaming platform with Amazon server that allows hosting, organization and distribution of all academic videos.
4. Anti-plagiarism service (plagiarism).
5. Subscribed databases (Digitalia Libraries and LIRN library network)
6. Training and extension platform (management skills school, language school, Scoop.it content curation network, and resource center).

**Table 1.** Structure of the departments involved in an academic facilitation.

| Department | Professionals | Number of Professionals |
|---|---|---|
| Educational Technology | Technology Director | 1 |
| | Technology Coordinator | 1 |
| | Developer | 1 |
| | Graphic Designer | 2 |
| | Layout Artist | 1 |
| | Audiovisual Technician | 2 |
| | Platform Coordinator | 1 |
| | Platform Manager | 2 |
| Teaching | Academic Director | 2 |
| | Academic Coordination | 3 |
| | Instructional Technicians | 2 |
| | Academic/Research | 56 |
| | Director of Research | 1 |
| University Wellness (BU) | BU Responsible | 1 |
| | BU Technicians | 2 |
| Registration and Control (R&C) | R&C Responsible | 1 |
| | R&C Technicians | 7 |
| Data Analytics | AD Responsible | 1 |
| | AD Technicians | 5 |
| Academic Mentoring | Academic Responsible | 1 |
| | Academic Mentors | 8 |

Own elaboration based on the information obtained by the HEI.

### 3.2. Activities Map

Once the inputs have been analyzed with the information offered by the different departments, the actions and tasks carried out in the different units are described. The result of this analysis has made it possible to group these actions and tasks into six activities: planning, design, development, teaching, mentoring/monitoring, and service. These activities are related to course outcomes and involves identifying and mapping each specific activity required to achieve learning objectives, allowing for better organization and coordination of the different stages of the online training process. Table 2 describes the tasks and actions related to each of the finalist outputs.

**Table 2.** List of activities associated with actions and tasks.

| Output Finalists | Actions and Tasks |
|---|---|
| PLANNING | Provision of the necessary infrastructure to facilitate access to courses in conditions of accessibility, usability, connectivity, etc. <br> Analysis of software and hardware acquisition needs associated with the course <br> Sequence training itineraries <br> Establish prerequisites and criteria for each course <br> Create virtual classrooms <br> Configure virtual classrooms <br> Establish rules for accessing and viewing courses <br> Configuring formulas and calculations associated with the course grade <br> Prepare program memories. |
| TEACHING | Review content with the academic area. <br> Review and update content <br> Order to library. <br> Invigorate the classroom. <br> Perform synchronous sessions. <br> Solve doubts. <br> Monitor tasks. <br> Advise academically. <br> Transfer research results. |
| MENTORING and FOLLOW-UP | Carry out the student reception process. <br> Continuously monitor student progress. <br> Establish regular communications with students <br> Establish the necessary actions to solve problems that arise to students. <br> Support students in resolving technical or administrative issues. <br> Establish communications with teachers. <br> Solve problems with teachers. <br> Support compliance with the academic calendar of each semester. |
| DESIGN | Delimitation of the objectives, contents, competences and learning results. <br> Design of curricular contents. <br> Analysis and validation of content quality. <br> Development of instructional design proposals. <br> Definition of activities associated with the content. <br> Planning of tools to be used in the course. <br> Scheduling synchronous encounters. <br> Structuring of learning units. <br> Definition of evaluation strategies |
| DEVELOPMENT | Coordinate with the academic area in the creation of resources. <br> Creation of multimedia content in Scorm format. <br> Creation of audiovisual material. <br> Content layout. <br> Evaluation of the accessibility and usability of the created resources. <br> Redesign of updated content. <br> Correcting, updating and republishing multimedia content. <br> Correcting, updating and republishing audiovisual content. |
| SERVICE | Advise students on administrative processes (registration, admission, graduation, etc.) <br> Carry out the registration, admission, transfer, postponement, disconnection, and graduation processes of students <br> Accompany the student in all the processes of continuous improvement of the institution. <br> Continuously attend the service of procedures and student communication. <br> Carry out the process of technical support and supervision of the online methodology for students. <br> Support in the digital training of all members of the HEI. <br> Development of welfare services for the university community |

Own elaboration based on the information obtained by the HEI.

### 3.3. Determination of Resource Cost Drivers

Once the activities of the HEI are known, resource consumption by the activity (cost driver) will be concluded and the percentage of the budget consumed by the affected departments within the HEI will be established, as described in Table 3.

**Table 3.** Resource inducers.

| Inputs | Code | Inductors | % Total |
|---|---|---|---|
| Salary Team Technology | ETEC | Hours worked | 10.1% |
| Academic Team Salary | EACA | Hours worked | 66% |
| University Welfare Team Salary (BU) | EBU | Hours worked | 2.8% |
| Registration and Control (R&C) Team Salary | ER&C | Hours worked | 4.0% |
| Data Analytics Team Salary | EAD | Hours worked | 3.3% |
| Salary of the Academic Mentoring Team | EMA | Hours worked | 7.9% |
| Technological Infrastructure (server architecture, LMS, SRM, etc.) | IT | Annual License | 4.0% |
| Educational Resources (library, subscribed databases, licenses, simulators, conference modules, proctoring, anti-plagiarism software, etc.) | RED | Annual License | 1.6% |
| Multimedia and Audiovisual Production Software | SFT | Annual License | 0.3% |

Own elaboration based on the training obtained by the HEI.

*3.4. Determination of Resource Cost Drivers*

The last part of the analysis involved establishing the percentage of the budget consumed by the activities necessary to develop the virtual programs (Table 4). For this, the first step required identifying the percentage of resources that are used in each activity (Figure 1).

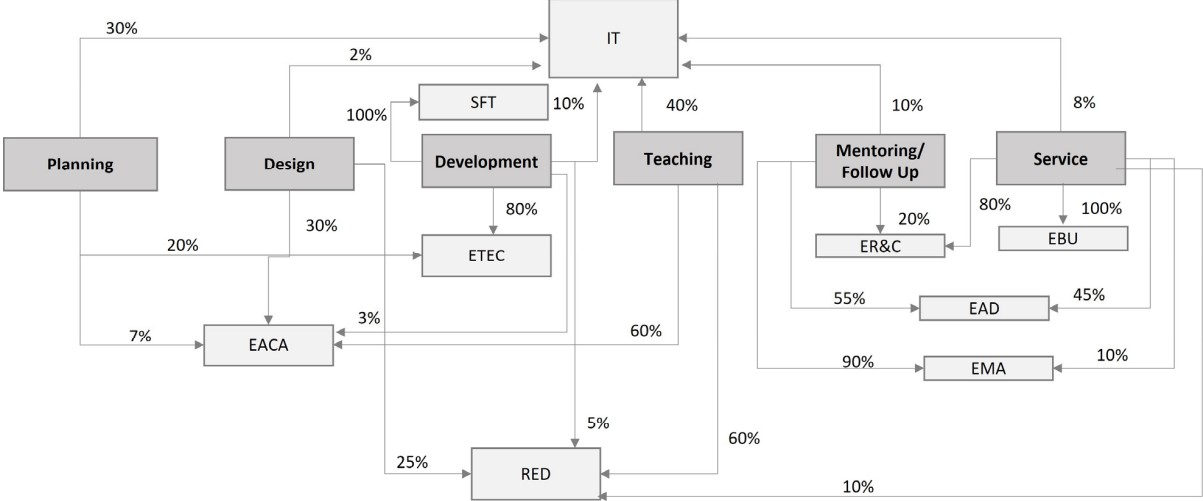

**Figure 1.** List of activities and resources.

Table 4 shows the costs that represent the resources assigned to specific activities, allowing us to see the impact that each of these activities has on the training process.

**Table 4.** Percentage of resource costs assigned to activities.

| Inputs/Outputs | Planning | Design | Development | Teaching | Mentoring/Follow Up | Service |
|---|---|---|---|---|---|---|
| ETEC | 2.0% | - | 8.1% | - | - | - |
| EACA | 4.6% | 19.8% | 2.0% | 39.6% | - | - |
| EBU | - | - | - | - | - | 2.8% |
| ER&C | - | - | - | - | 0.8% | 3.2% |
| EAD | - | - | - | - | 1.8% | 1.5% |
| EMA | - | - | - | - | 7.1% | 0.8% |
| IT | 1.2% | 0.1% | 0.4% | 1.6% | 0.4% | 0.3% |
| RED | - | 0.4% | 0.1% | 1.0% | - | 0.2% |
| SFT | - | 0.03% | 0.3% | - | - | - |
| % TOTAL | 7.8% | 20.3% | 10.8% | 42.2% | 10.1% | 8.8% |

Note: Own elaboration based on the information obtained by the HEI.

As it can be noticed in the table of assigned costs, teaching is the activity that consumes the vast amount of resources within the HEI, followed by the design of training actions. Along with these, student follow-up activities and the provision of adequate educational services consume 81.4% of the assigned resources, revealing that only 18.6% of the resources are used in essential activities of the virtual modality, such as planning and configuration of virtual classrooms, and using necessary technology for the training process and the development of multimedia courses.

## 4. Discussion

The analysis carried out in this study has allowed to broaden the knowledge of the cost structure and activities of an HEI that offers training in virtual modality. The model designed has been established as a source of useful information to know the costs associated with each of the activities of the institution and offer the opportunity to assess strategies that can be oriented towards improving the quality and efficiency of each one of these functions.

The results obtained from the analysis show how the teaching activities associated with the design of the training processes are generating a high cost in the total budget of the departments, coinciding with previous studies, such as those carried out by Doberti [54] that determined how labor continues to be one of the most important cost factors in virtual teaching modes, representing approximately 75% of the total budget.

Along with the above, it is verified that in virtual learning, there is a high level of importance that the presence of other professionals acquires, who have adequate skills to carry out tasks that requires a more intense use of technology, and who can develop activities associated with this modality, such as instructional design, multimedia and audiovisual production or the configuration of virtual classrooms, among others.

The multiplicity of professional profiles with skills and technological training associated with training in virtual modality contradicts the conclusions of research, such as those developed by Deming et al. [2] or Ortagus and Yang [13]), among others, that highlighted the reduction in labor costs associated with teaching as one of the advantages of virtual education, justifying them based on an economic model of scale where the initial investment in course production was later offset by a higher enrollment fee and less teacher interaction.

Keeping in mind the previous conclusions and emphasizing this reduction in labor costs associated with virtuality, it should be noted that each educational system has its own protection mechanisms and guarantees so that, regardless of the modality adopted, students have adequate educational accompaniment. In this present study, the HEI is recognized as an educational service provider within the Colombian territory and as such is obliged to comply with the rules regarding the sufficiency of the teaching staff and report to the National Higher Education Information System (SNIES, acronym in Spanish for *Sistema Nacional de Información de la Educación Superior*) This guarantees the presence of the sufficient number of teachers who accompany the students in each of the programs and therefore invalidates the possibility of reducing costs associated with the modality by increasing the number of enrollments for each assigned teacher.

On the other hand, cost reduction associated with economies of scale in the production of courses, as defended by Ortagus and Yang [13], is also an issue that must be considered with caution. Authors, such as Segovia-García and Said-Hung [33] point out how the obsolescence of the contents arranged to generate training actions directly influences the perception of quality and satisfaction of students with their program and institution, resulting in demotivation and decline in academic performance. In this regard, as observed in the activity's matrix, the production of materials associated with training programs and actions is not a one-time activity, but rather requires continuous improvement and updating actions and therefore the use of resources and activities associated with the budget. This continuous process of improvement is closely linked to different models of instructional design, such as ADDIE, wherein the logical sequence of work evaluation is contemplated as a final phase but also in the rest of the phases too (analysis, design, development and

implementation), a holistic approach is adopted in identifying and continuously reviewing the quality criteria and improving actions to review and update these materials.

The work presented here has made it possible to directly analyze the activities and costs generated in the area of academic service provision of an HEI that works in virtual mode. Additionally, although the main objective of the study has not been to determine if education in virtual modality is cheaper than face-to-face, it has been corroborated that, when virtual training is carried out at a large scale, the development costs become significant [14].

Future studies should expand the scope of study that would allow a comparison of the costs generated for the same program in different training modalities: virtual, face-to-face, and mixed. This would to generate and expand knowledge on the cost of activities in each modality, which will eventually provide strategic information to make correct management decisions.

## 5. Conclusions

This study probes into the cost and activity structure of an HEI that offers virtual education using a model, ABC, which allows for a better understanding of how resources and costs are distributed in the different branches of educational services. From the results obtained, the following conclusions can be highlighted:

- The designed model allows to acquire knowledge on the costs associated with each activity and facilitates the evaluation of quality and efficiency improvement strategies, as well as aids in the optimization of decision-making processes within institutions.
- Among the analyzed activities, teaching generates the highest cost within the total budget of the departments associated with virtual education.
- Other activities, such as instructional design, multimedia and audiovisual production, and virtual classroom configuration, are also important in imparting virtual education.
- The multiplicity of professional profiles with technological competencies required in this field contradicts the idea of reducing labor costs associated with virtual teaching and highlights the importance of human capital.
- The cost reduction associated with economies of scale in course production must be considered with caution, as the obsolescence of designed content or materials can influence the perception of quality and student satisfaction, thus discrediting the virtual methodology.
- The continuous improvement process in material production requires resources and activities associated with the budget, and therefore virtual education can be significantly expensive when executed at a large scale.

**Author Contributions:** Conceptualization, N.S.-G. and E.M.-C.; methodology, N.S.-G.; software, N.S.-G. and E.M.-C.; validation, E.M.-C.; formal analysis, N.S.-G.; investigation, N.S.-G. and E.M.-C.; resources, N.S.-G.; data curation, N.S.-G. and E.M.-C.; writing—original draft preparation, N.S.-G.; writing—review and editing, N.S.-G.; supervision, E.M.-C.; project administration, E.M.-C. All authors have read and agreed to the published version of the manuscript.

**Funding:** This research received no external funding, and the APC was funded by Corporación Universitaria de Asturias.

**Institutional Review Board Statement:** Not applicable.

**Informed Consent Statement:** Not applicable.

**Data Availability Statement:** The data is available at: https://cutt.ly/F6PCz6F (accessed on 9 April 2023).

**Conflicts of Interest:** The authors declare no conflict of interest.

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
