# Peer review of "Cost Analysis in Online Teaching Using an Activity Map"

_education, doi:10.3390/educsci13050506_

Round 1
Reviewer 1 Report
Appreciate the authors for their study and approach of analysis of the costs of a virtual university using an ABC activity model to determine the percentage of the budget consumed by each of the activities involved in the training process. Having said that the authors are to clarify the below-given comments point by point.
Q1. The introduction is well-written. A para at the end of the introduction is needed to state the present scope of work in connection to the study
Q2. The authors have mentioned in the abstract the importance of weight for instructional design and teaching activities to maintain in educational modalities mediated by technology. By what % of the weight do you think the present study would give to the above?
Q3. In the materials and method section, the authors have mentioned that the analysis has been temporarily limited to the period from January to December 2021. How do you justify the relevance of the study post-Covid-19?
Q4. I feel the curriculum is missing in Table 1, How do you compensate that in the study?
Q5. In Table 2, how do you map the list of activities with the course outcomes and learning objectives?
Q6. The discussion section is good, I suggest the authors may list the highlights of the study in a separate section. conclusions/summary
Q7. How do you suggest the present analysis relavant to other state Universities otherthan Columbian higher education? Mention it in a seperate section "scope of the work"
Author Response
Dear doctor, we appreciate the review conducted on the work presented. We have addressed all the issues pointed out in your evaluation and resubmitted the work. In the text, we have highlighted the changes made in red. We attach the file with the changes and then identify each one of them.
|
|
|
|
|
|
5 |
1.2. Scope of work. |
Q1. The introduction is well-written. A para at the end of the introduction is needed to state the present scope of work in connection to the study |
The last paragraph of the introduction has been reworked, creating a section on the Scope of the Study where the scope of the study has been detailed. |
|
1 |
ABSTRACT |
Q2. The authors have mentioned in the abstract the importance of weight for instructional design and teaching activities to maintain in educational modalities mediated by technology. By what % of the weight do you think the present study would give to the above? |
Percentage of activities related to teaching and instructional design tasks has been included in the abstract |
|
5 |
1.2. Scope of work. |
Q3. In the materials and method section, the authors have mentioned that the analysis has been temporarily limited to the period from January to December 2021. How do you justify the relevance of the study post-Covid-19? |
In the scope section of the study, it has been explained that events such as COVID do not have an impact on this type of study where the reference institution is 100% virtual and has not changed either its way of working during COVID or its tools |
|
5 |
1.2. Scope of work. |
Q4. I feel the curriculum is missing in Table 1, How do you compensate that in the study? |
Also in this section of Scope of the study, it has been indicated that the activities analyzed are those after the design of study plans, and therefore, the curriculum section does not affect the study. |
|
8 |
3.2 Activity maps |
Q5. In Table 2, how do you map the list of activities with the course outcomes and learning objectives? |
The text has indicated how the activity map relates to course outcomes and learning objectives by identifying and mapping each specific activity necessary to achieve the learning objectives. |
|
12 |
5. Conclusions |
Q6. The discussion section is good, I suggest the authors may list the highlights of the study in a separate section. conclusions/summary |
Following the reviewer's recommendations, a conclusion section has been included with the highlights of the study. |
|
5-6 |
1.2. Scope of work. |
Q7. How do you suggest the present analysis relavant to other state Universities otherthan Columbian higher education? Mention it in a seperate section "scope of the work" |
In the "scope of the work" section, it has been described how the study has been designed and its possible adoption by other universities to analyze their costs |

Reviewer 2 Report
Thanks for providing me with the opportunity to read the manuscript. The suggestions are:
1) Please clarify the ABC activity model. Usually, it needs to be clearly defined if it is the first time to be presented in the abstract.
2) Activity Map is in the title. But I do not know what it is. Does it mean the activities that the teacher(s) will map it out?
Thanks for providing me with the opportunity to read the manuscript. The suggestions are:
1) Please clarify the ABC activity model. Usually, it needs to be clearly defined if it is the first time to be presented in the abstract.
2) Activity Map is in the title. But I do not know what it is. Does it mean the activities that the teacher(s) will map it out?
Author Response
Dear doctor, we appreciate the review conducted on the work presented. We have addressed all the issues pointed out in your evaluation and resubmitted the work. In the text, we have highlighted the changes made in red. We attach the file with the changes and then identify each one of them.
|
|
|
|
|
|
1 |
ABSTRACT |
Q1. Please clarify the ABC activity model. Usually, it needs to be clearly defined if it is the first time to be presented in the abstract.
|
It has been clarified in the abstract that the ABC model stands for Activity-Based Costing. |
|
1 |
ABSTRACT |
Q2. Activity Map is in the title. But I do not know what it is. Does it mean the activities that the teacher(s) will map it out? |
A study scope section has been included explaining the importance of the activity map within the methodology |

Round 2
Reviewer 2 Report
Please polish it and then it is ready to be published.
Please polish it and then it is ready to be published.
Author Response
Thank you again for your interest in our work, and we would like to inform you that we have just uploaded it, revised and complete.